# Mother–Child Approach to Cervical Cancer Prevention in a Low Resource Setting: The Cameroon Baptist Convention Health Services Story [†]

Lorraine Elit [1,*] , Florence Manjuh [2], Lillian Kila [3], Beatrice Suika [2], Manuela Sinou [4], Eliane Bozy [4], Ethel Vernyuy [4], Amandine Fokou [4], Edith Welty [5] and Thomas Welty [5]

[1]  Cameroon Baptist Convention Health Services: Mbingo Site, Mbingo, Cameroon
[2]  Cameroon Baptist Convention Health Services: Etougebe Site, Yaounde, Cameroon; manjuhrencefav@yahoo.com (F.M.); suikangoran6@gmail.com (B.S.)
[3]  Cameroon Baptist Convention Health Services: Banso Site, Banso, Cameroon; kilalilian7@gmail.com
[4]  Cameroon Baptist Convention Health Services: Meskine Site, Meskine, Cameroon; siwaima@yahoo.fr (M.S.); elianebozi73@gmail.com (E.B.); lovemyuy@gmail.com (E.V.); fokouamandine32@gmail.com (A.F.)
[5]  Cameroon Baptist Convention Health Services: Nkwen Site, Bamenda, Cameroon; ediewelty@gmail.com (E.W.); thomaswelty@gmail.com (T.W.)
*  Correspondence: elitl@mcmaster.ca
[†]  Presented at the 11th Annual International Cervical Cancer Prevention Conference, Dschang, Cameroon, 21 March 2024.

**Abstract:** Introduction: The rates of cervical cancer screening in Cameroon are unknown and HPV vaccination coverage for age-appropriate youths is reported at 5%. Objectives: To implement the mother–child approach to cervical cancer prevention (cervical screening by HPV testing for mothers and HPV vaccination for daughters) in Meskine, Far North, Cameroon. Methods: After the sensitization of the Meskine–Maroua region using education and a press-release by the Minister of Public Health, a 5-day mother–child campaign took place at Meskine Baptist Hospital. The Ampfire HPV Testing was free for 500 women and vaccination was free for age-appropriate children through the EPI program. Nurses trained in cervical cancer education conducted group teaching sessions prior to having each woman retrieve a personal sample. Self-collected samples were analyzed for HPV the same day. All women with positive tests were assessed using VIA–VILI and treated as appropriate for precancers. Results: 505 women were screened, and 92 children vaccinated (34 boys and 58 girls). Of those screened, 401 (79.4%) were aged 30–49 years old; 415 (82%) married; 348 (69%) no education. Of the HPV positive cases (101): 9 (5.9%) were HPV 16, 11 (10.1%) HPV 18, 74 (73%) HPV of 13 other types. Those who were both HPV and VIA–VILI positive were treated by thermal ablation (63%) or LEEP (25%). Conclusion: The mother–child approach is an excellent method to maximize primary and secondary prevention against cervical cancer.

**Keywords:** cervical cancer prevention; HPV vaccination; HPV testing

## 1. Introduction

Globally, cervical cancer ranks fourth in incidence of cancer and fourth in mortality from cancer. Cervical cancer affected 604,000 women globally with 342,000 deaths in 2020 [1]. Ninety percent of cervical cancers occur in low- and middle-income countries with 19 of the top 20 countries located in sub-Saharan Africa [2].

Cameroon has a population of 28 million with 8 million women aged 15 years and older. Every year 2770 women are diagnosed with cervical cancer and 1787 die from disease [3]. This is likely an under-reporting of cases given the lack of a national cancer registry in the country. In Cameroon, cervical cancer is the second leading cause of cancer and cancer deaths for women.

Cervical cancer is a preventable cancer. An initial screening using the Papanicolaou test has been around for 75 years. The use of Visual Inspection with Acetic Acid (VIA) followed by Lugol's iodine (VILI) has been a low-cost alternative for low and middle resource countries in women 30 years and older. The sensitivity is roughly equal when comparing both tests as a single screening. More recently, women using a self-sampling device to collect cells or physician-directed testing to obtain cells that are then tested for HPV is a highly sensitive screening test (more than 90%). However, a positive result for high risk HPV type(s) mainly indicate current infection. Women must undergo a second test like VIA–VILI to determine if there is infection and dysplasia present.

Primary prevention is available using the HPV vaccine. Either the quadrivalent or nonavalent vaccine provide excellent protection in girls and boys aged 9–14 years as shown by the durability of the three doses, two doses and more recently one dose schedules [4]. This prevention strategy is effective against HPV-related cervical, vaginal, vulvar, anal, and oropharyngeal cancers.

In Cameroon, the HPV vaccine was introduced in 2010 through a donation to the Cameroon Baptist Convention (CBCHS) Women's Health Program (WHP) of quadrivalent Gardasil [5]. Considering the success of the pilot program, the Cameroon Ministry of Health (MOH) applied and received subsidized HPV vaccine from GAVI and provided quadrivalent Gardasil to the WHP at no charge. In 2021, the HPV vaccination coverage of 9-year-old girls was estimated at 20% [3]. Coverage for cervical cancer screening is estimated to be 5% for ever screened women ages 25–65 years and 6% for those 30–49 years, (4% and 5% for those screened in the last 5 years and 3% and 4% for those screened within the last 3 years, respectively) [3]. Thus, in the low- and middle-income settings like Cameroon, the issue is now to get vaccine doses into the arms of 9–14-year-old girls and boys and provide cervical cancer screening to women aged 30 and above.

*Uniqueness of the Mother–Daughter Approach [6]*

The mother–daughter or mother–child approach to improving cervical screening and HPV vaccination rates has been conceived and implemented successfully by the CBCHS WHP [7,8]. The mother–child approach to date has been implemented using a mass campaign approach. The concept involves a buy-in from local health and religious authorities. Thereafter, there is the training of healthcare workers concerning cervical cancer and ways to prevent disease. CBCHS health workers use various strategies to provide community sensitization in schools, churches, and village meeting areas in the location of focus. The sensitization is followed by a campaign for screening and vaccination. The dates are advertised. The campaign is held at a local health facility. These campaigns have to date been funded by grants, so are free of charge to the mothers for screening, and the vaccination is free of charge through the countries EPI supply.

## 2. Methods

Meskine is a suburb of Maroua, which is the capital of the Far North Region of Cameroon. The region is bordered by Chad and Nigeria. Meskine has a population of 16,000 while that of Maroua is more than 200,000. Meskine is known for its market garden crops like tomatoes, onions, and millet [9]. The area is made up of the various ethnic groups Guiziga, Moufou, Guidars and Peuls. In 2014, the poverty rate was 74.3% compared to the national average of 37.3%. Literacy in this region is 39.8% for the women and higher for men [10,11].

Hôpital de Meskine (or Meskine Baptist Hospital Maroua) is a tertiary care facility located in the far north of Cameroon. It has a 125-bed capacity. There is currently an on-site gynecologist, general surgeon, internist and several house officers. This facility was previously (1993–2019) owned and operated by the Medical Centers of West Africa (Centre Medicaux de L'Afrique de L'Ouest CMAO). Following the Boko Haran insurgence, the missionaries left in 2014. In late 2019, the CBCHS agreed to take over the management of the facility. The WHP of CBCHS agreed to make Hopital de Meskine its 14th WHP

site. A family planning nurse leads the program that currently exists. The director and another nurse were trained in cervical cancer screening using the Ampfire HPV testing, VIA–VILLI screening, and biopsy, thermocoagulation or LEEP treatment. Higher levels of patient acuity require referral to specialist care.

### 2.1. Cervical Cancer Screening

A fixed obligation grant came through the Charities Aid Foundation of America to cover the cost of breast and cervical cancer screening for 500 women at Meskine Baptist Hospital. The Minister of Public Health (Dr. Manaoudu Malachie) published a press release on 12 January 2024, concerning the campaign. WHP provided education for the community and religious leaders, which included the sensitization of local churches and mosques, a school, and village meeting centers (see Figure 1). In addition to local staff, the WHP program director (from Etoug-Ebe Baptist Hospital), an experienced WHP nurse fluent in the Hausa language (from Banso Baptist Hospital), a lab technician who could run the HPV test (from Etoug-Ebe Baptist Hospital) and a gynecologic oncologist (from Mbingo Baptist Hospital) joined the program launch from 22–26 January 2024.

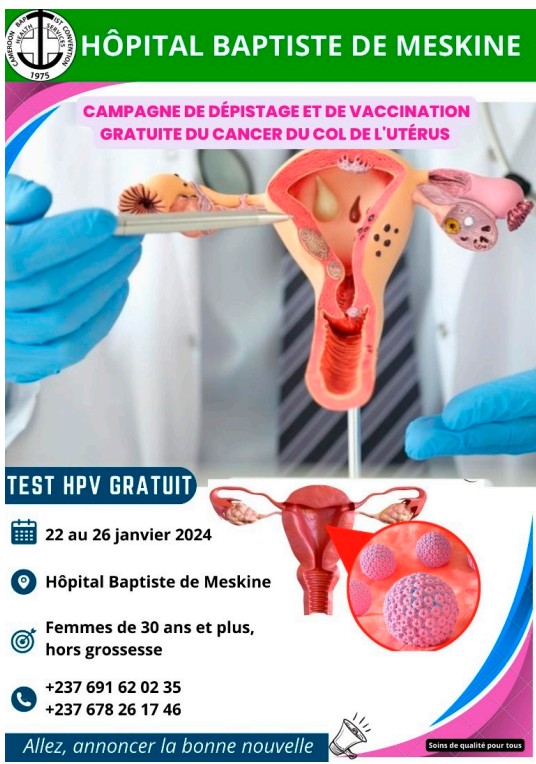

**Figure 1.** Handout concerning the campaign. Part of the sensitization approach.

All women who attended the campaign received a number and either brought or purchased a small book that they would retain containing their name, contact information and personal test results.

The women received group or individual education about cervical and breast cancer, about the potential to prevent cervical cancer through screening or vaccination, and education on how a woman uses a self-sampling device to obtain cells that are then tested for HPV (Figures 2 and 3). Every woman was interviewed in her language and a four-page registration form was completed which included information on the woman's age, village, at least two phone numbers, history of breast or cervical screening, history of breast or cervical cancer symptoms, HIV status and most recent testing date and medication use. All women underwent nurse-led breast palpation. Any women with breast lesions were referred for either ultrasound or consultation.

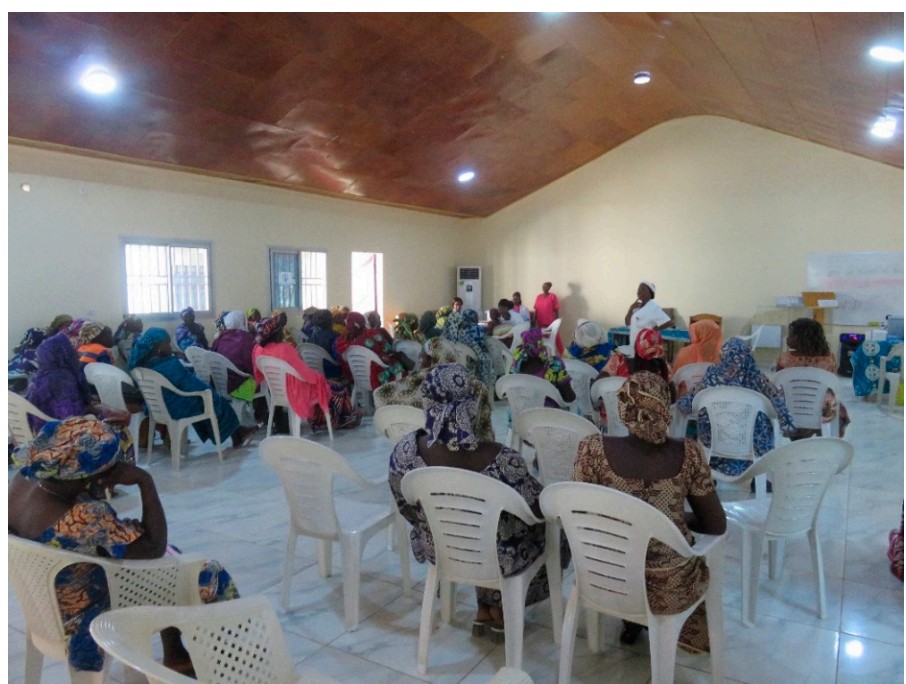

**Figure 2.** Group Health education.

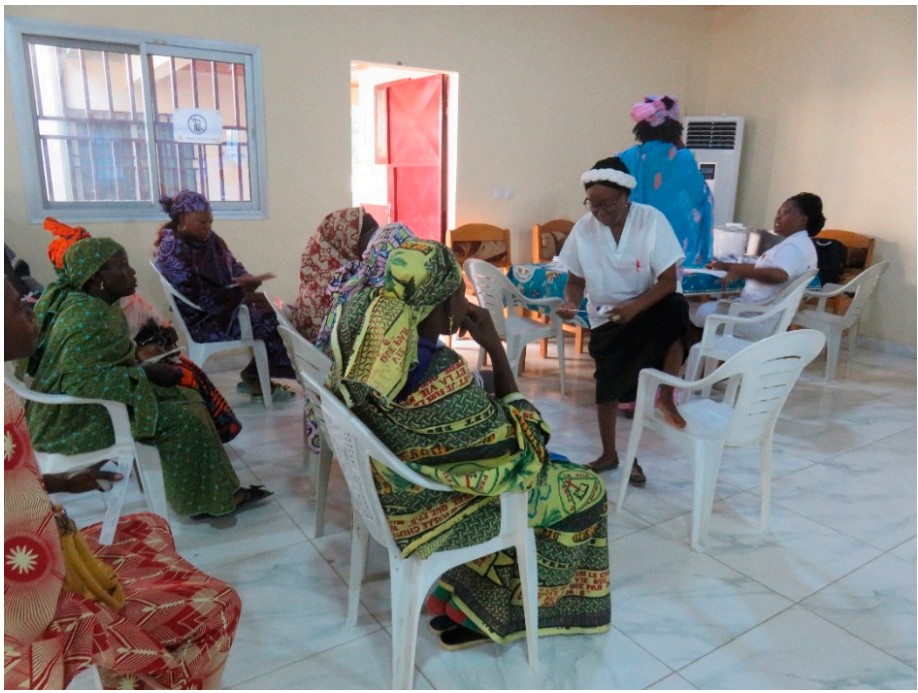

**Figure 3.** Demonstrating the HPV Self-sampling.

HPV screening used the AmpFire HPV analyzer (Figure 4). The Ampfire is a simple and fast HPV detection technology using a multiplex isothermal real-time fluorescent detection assay. This novel HPV screening technology can detect types 16, 18 and 13 other high risk HPV types. The time from sample to result is one hour twenty minutes. The machine was transported from Etoug-Ebe Baptist Hospital by plane. Two or three runs of up to 94 samples were conducted each day. Women tended to wait for their results and complete their designated management strategy same day.

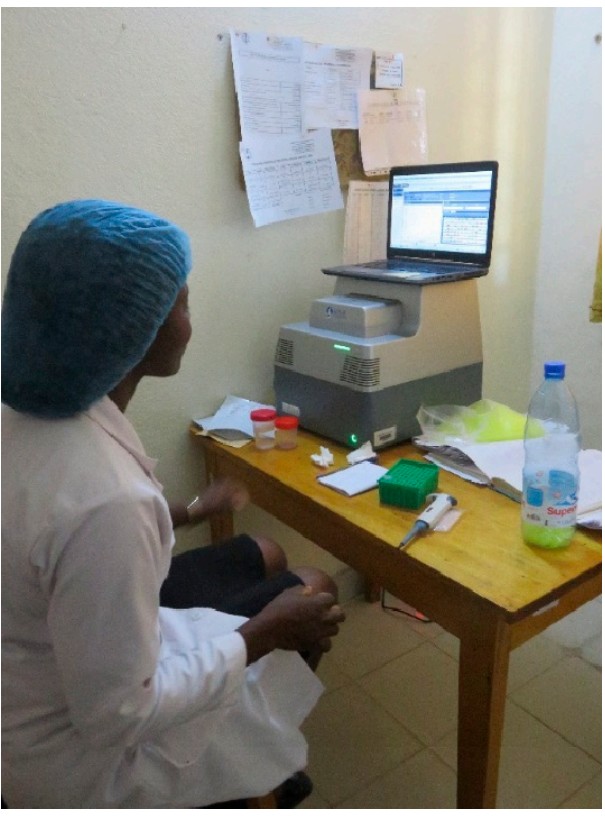

**Figure 4.** Waiting for HPV test results from the Ampfire machine.

The CBCHS WHP provided VIA–VILI enhanced by Digital Cervicography (DC) for those women who were positive for oncogenic HPV or who presented with symptoms. Women with precancerous lesions who met the World Health Organization (WHO) criteria for ablation were treated by thermocoagulation. Women with lesions extending into the endocervical canal underwent LEEP. Costs up to this level of care were covered. Women with cancer were assessed and referred as appropriate.

*2.2. HPV Vaccination*

Any age-appropriate child (male or female, 9–14 years old) with parental/guardian verbal consent was vaccinated with the quadrivalent Gardasil vaccine provided through the EPI program at Meskine Baptist Hospital. The children were observed for 20 min post vaccination. The cold chain was preserved. All children were provided with a blue vaccine card indicating their name, the date, place, and type of vaccine given. All names, ages, genders, telephone numbers, addresses, and parent names were record in the log as is the standard of care for vaccination.

*2.3. Ethics*

The final manuscript was reviewed by the CBCHS Institutional Ethics Board.

**3. Results**

*3.1. Cervical Screening*

There were 505 women who were screened. The woman who travelled the farthest made a 2 day journey one way from Chad. The campaign was for women 30 and above; however, 9 younger women were seen and so were screened with VIA alone (Table 1). Most of the women were married (82%); 79% of women were 30–49 years old. Many women (69%) had had no education. Half of the women were Christian and a third were Muslim. Less than 1% had known positive HIV status but 48% did not know their status.

**Table 1.** Demographics.

| Category | Number | Percent |
| --- | --- | --- |
| Age (years) | | |
| 20–29 | 9 | 1.8 |
| 30–49 | 401 | 79.4 |
| 50–69 | 83 | 16.4 |
| 70+ | 3 | 0.6 |
| Unknown | 9 | 1.8 |
| Marital Status | | |
| Married | 415 | 82.2 |
| Single | 14 | 2.8 |
| Divorced | 16 | 3.2 |
| Separated | 7 | 1.4 |
| Widowed | 44 | 8.7 |
| Unknown | 9 | 1.8 |
| Education | | |
| None | 348 | 68.9 |
| Primary | 96 | 19.0 |
| Secondary | 29 | 5.7 |
| High school | 14 | 2.8 |
| Higher education | 18 | 3.67 |
| Religion | | |
| Christian | 267 | 52.9 |
| Baptist | 47 | |
| Catholic | 66 | |
| Presbyterian | 67 | |
| Pentecostal | 87 | |
| Muslim | 164 | 32.5 |
| Other | 58 | 11.5 |
| None | 16 | 3.2 |
| HIV status | | |
| Negative | 255 | 50.5 |
| Positive | <5 | |
| Unknown | 246 | 48.7 |

505 women completed HPV tests. The HPV test was negative in 404 (80%). Of the 101 positive HPV tests (20%): there were 9 with HPV 16 (9%), 11 with HPV 18 (10.1%), 74 (73%) with HPV other 13 high risk types. Seven percent had more than one oncHPV type: 3 with HPV 16 and HPV of 13 other types (3.0%), 2 with HPV 18 and HPV of 13 other types (2.0%), 1 with HPV 16 and 18 (1.0%), 1 with HPV 16, 18 and 13 other HR types (1.0%).

All HR HPV positive cases underwent VIA and VILI (Figures 5–7) except for one patient who was identified as being pregnant on further questioning. During further assessment, two women were found to be symptomatic with months of continuous bleeding and foul discharge and were presented for diagnosis and not screening. Both women had advanced stage cervical cancer (at least Stage 3c) on clinical examination. Confirmatory histology, imaging, extensive counselling, and referral for chemo-radiation therapy occurred.

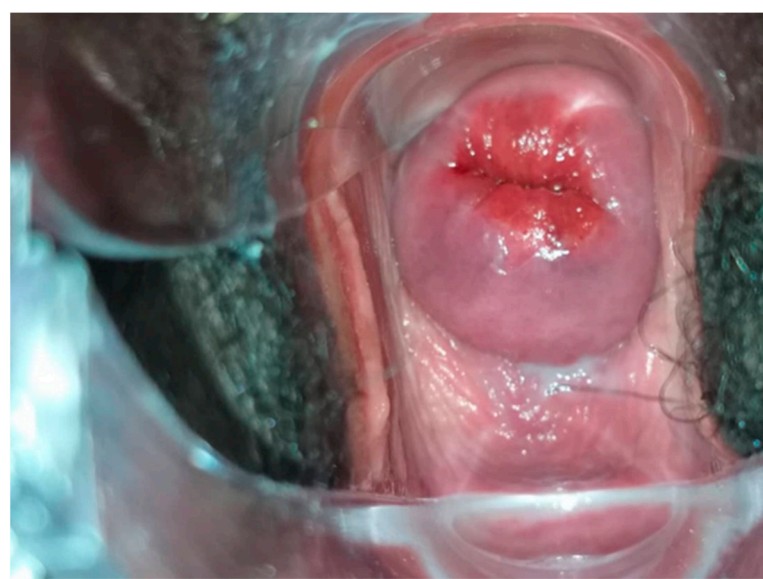

**Figure 5.** HPV positive Cervix with no magnification.

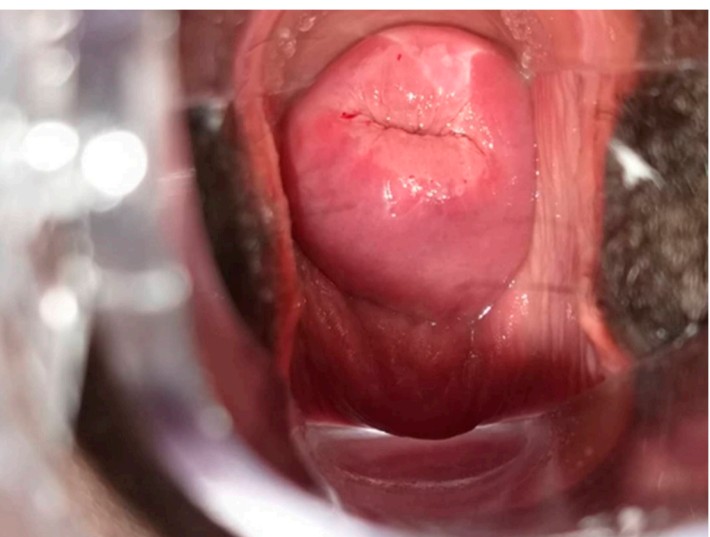

**Figure 6.** Figure 5 cervix after acetic acid (VIA). Considered VIA positive.

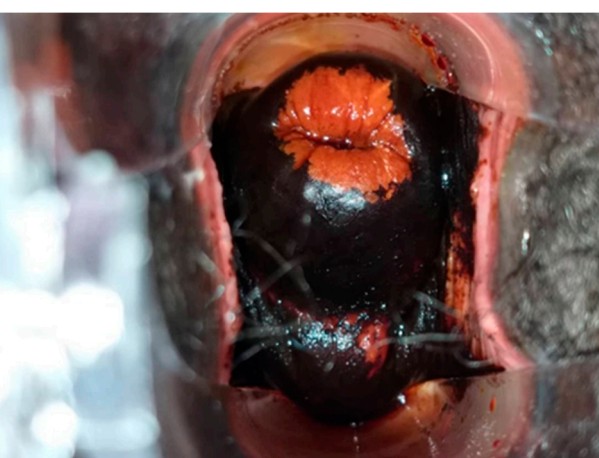

**Figure 7.** Figure 5 cervix after Lugol's iodine (VILI). Considered VILI positive.

There were eight women who were HR HPV positive and VIA–VILI positive (8%). Five women (62.5%) underwent thermocoagulation. Two women (25%) underwent the LEEP procedure. One woman had an active pelvic inflammatory disease and was treated with antibiotics with plans to return for treatment in 2 weeks.

There were three invalid cases that could not be ratified by a repeat run of the original specimen through the Ampfire machine or a repeat self-sampling testing. These women were screened by VIA–VILI.

### 3.2. HPV Vaccination

Prior to this campaign, the quadrivalent vaccine was available at the hospital, but no doses had been given. There was a sensitization of only one school, as the hospital had not received the authority to sensitize more widely. During the week of the WHP launch, 92 children were vaccinated (34 boys and 58 girls). These children tended to come before or after classes. The number of children attending for vaccination increased the longer the campaign. There were no acute toxicities aside from pain at the injection site. One delightful story was that of a grandmother who had come for screening the day prior and brought her seven age-appropriate grandchildren the following day for vaccination. The HPV vaccine will remain available free of charge through the Meskine Hospital EPI program after the culmination of the program launch.

### 3.3. Breast Screening

All women were to receive breast screening by palpation. There were six women with palpable masses. They underwent ultrasound and, if appropriate, biopsy and follow-up through the gynecology or surgery programs.

## 4. Discussion

We report on a relatively novel strategy for cervical cancer prevention involving mothers and their children. To put the value of this strategy into context, we will discuss first why the uptake of cervical screening and vaccination is so low. Even though the MOH has a good track record with childhood immunization, the population has been hesitant in receiving the HPV vaccines. Since January 2023, the quadrivalent vaccine has been free for girls and boys aged 9–14 through the EPI program in Cameroon. The vaccine uptake is low for a host of reasons: lack of adequate sensitization [4,12], misinformation on social media [4], the recent COVID-19 pandemic [4,12], and/or lack of trust in the government-based introduction of new vaccines in this decade [12], guardians of children not aware of the vaccine or reasons for the vaccine [12], and its effectiveness or its availability [12]. This is compounded in rural locations of much of the population or residence in regions of conflict [12]. The MOH has implemented both clinic-based and school-based approaches and has collaborated with the Ministry of Education. However, neither of these approaches have been productive. The MOH has called on private and faith-based healthcare organizations to assist them in administering the vaccines [5].

The cost of medical care in Cameroon is paid by patient fee. Thus, cervical screening by various techniques is associated with a cost to the client. Usually HPV testing (rural 7500 cfs to urban 10,000 cfs (USD 8.26–16.53)) is the highest cost test with VIA VILI (rural 3000 cfs to urban 5000 cfs (USD 4.94–8.26)) as the lowest. Any biopsies or treatments increase the patient bill. In a setting of poverty or conflict, the family resources are directed firstly to high priority needs like home and shelter, followed by education for children. Other issues include: limited resources, health infrastructure, shortage of healthcare providers, low level of awareness, and insufficient attention to women's health, especially in rural areas [13].

To increase vaccination rates in LMICs, a review was conducted of 21 HPV vaccination programs (using the 3-dose strategy) in LMICs which included Cameroon. Vaccine uptake rates were superior if there was community involvement, school and health clinic implementation models which were superior to other strategies, NGO versus other private management of the program, and a longer duration of the vaccination program [14].

The literature addresses effective ways to optimize cervical screening attendance in LMICs. In 24 studies (15 RCT and 9 cluster with RCTs including 1 study from Cameroon), self-sampling was superior to VIA (RR 1.93 95% CI 1.66–2.25); reminder phone calls or SMS texts facilitated attendance when compared to letter invitation (RR 1.72, 95%CI 1.27–2.32, RR 1.59, 95% CI 1.46–5.19); free or subsidized costs optimized attendance (RR 1.60, 95% CI 1.10–2.33); and community-based HPV testing was superior to hospital-collected HPV testing (RR 1.67, 95% CI 1.53–1.82) [15]. A recent sequential prospective study in the West Region of Cameroon showed that in the 10 months from September 2018–May 2019, there was a passive approach of community information channels with media announcements, posters in women's associations, churches and integrated health centers [16]. The subsequent nine months from June 2019 to February 2020 involved the recruitment and training of community health workers who spread the message. They received a USD 1.00 incentive for each woman recruited. The focus was on women 30–49 years old. The passive strategy recruited well initially from the urban setting of Dschang but the CHW worked best in the rural or hard-to-reach areas. It is important to note that 45% of Cameroon's population live in rural areas.

This is the first publication describing a mother–child approach to cervical cancer prevention by using a single dose HPV vaccination in children (age 9–14 years old), in combination with education and an oncHPV test cervical screening for mothers/guardians. There has been a previous study of this strategy in Ngounso Baptist Health Center in the West Region (Muslim dominant population) and Voundou Baptist Health Center (Christian dominant population) in the Centre Region (Table 2) [16]. In that project report, the Health Center staff and Community Health Workers (CHWs) from the CBCHS WHP were trained over 2 days on cervical cancer screening sensitization and HPV vaccination in 2021. The training had a session for role play on health education, cervical cancer screening and vaccination by participants while receiving feedback from facilitators. The topics that were discussed in the training included: an overview of the TogetHER project, basics of the natural history of cervical cancer, the basics of cervical cancer screening, HPV vaccination, and communication for behavioral change. The importance of adequate data collection and documentation was also emphasized. In both sites, there was buy-in from the government, local, and religious authorities through courtesy and advocacy visits. The number vaccinated were much higher than in our study. The HR HPV positivity rate was in keeping with that of Voundou (another Christian dominant population). We used aspects of that work in the current Meskine program.

**Table 2.** Current results of Mother–child prevention programs in Cameroon.

| | Nguonso [16] | Voundou [16] | Meskine |
|---|---|---|---|
| | Muslim dominant | Christian dominant | 50% Christian and 30% Muslim |
| Screened | 317 | 439 | 505 |
| HR HPV+ | 87 (2.5%) | 119 (27.1%) | 101 (20%) |
| VIA+ | 5 (5.7%) | 2 (1.7%) | 8 |
| Thermocoagulation | 5 (100%) | 2 (100%) | 5 (62.5%) |
| LEEP | | | 2 (25%) |
| Pending treatment | | | 1 |
| Diagnosis not screening Suspicious for cancer | 2 | | 2 |
| Vaccination Baseline—first dose | 911 | 981 | 92 |
| 6 mos—second dose | 308 | 283 | |

There is currently a proposal by Manga to conduct another campaign in two urban slums, one in the capital city, Yaoundé, and the other in the economic capital, Douala. No cervical screenings or vaccinations are currently available in these communities [16].

*Uniqueness and Challenges of the Meskine Campaign*

As Manga showed previously [16], the mother–child campaign increases vaccine uptake. This was seen in the Meskine campaign, but the numbers vaccinated were much smaller than in the Manga work [16].

The Meskine campaign population was 50% Christian and 30% Muslim. Roughly 70% of women had no education, which is in keeping with prior reports concerning this region [10,11]. This low literacy level provided a challenge in implementing a campaign. Women were each given a number when they arrived. Many women could not read and/or they did not know their numbers. Some women did not even know their names as written on their personal medical book.

The campaign was open from 7 am–6 pm and campaign staff had multiple tasks. The campaign facilitated the availability of a French speaking staff but at times, individuals were not available to speak the attendees' local dialect.

Those women with invalid tests had to remain at the clinic for extended periods of time for the test to be rerun or retaken and run and so as to receive a management plan.

On one of the five days, the Ampfire machine had cycled for 1 h and 10 min when the hospital experienced a power interruption. Once the power returned, the whole cycle had to begin again. Battery back up to the machine would be ideal.

In the Meskine situation, the Ampfire machine was transported by air to Meskine at significant cost. The machine was felt to be too sensitive to transport 3 days one-way over non-paved roads. The strategy Manga [16] used was that an individual travelled at night to transport specimens and returned the next day with results. This was not feasible for our large campaign and the large distances. However, in the future, from specimens collected in Meskine and sent to a central lab, results will likely be available in 2 weeks.

While the organizers planned for 120 and later 500 attendees, more women came than expected. Since the funding was only for 500, these additional women were offered testing at their own cost. We found that many women were willing to pay which is an important lesson moving forward, since screening at the hospital will be at the clients' expense.

Some women who attended the screening campaign did use the hospital outpatient services for other consultations. Manga also showed in his work that sensitization at schools, churches, and community meeting groups increased the relevance of the local health clinic [16].

Strengths of this mother-child approach include the pre-campaign sensitization including involvement of the Minister of Public Health which generated allot of com-munity discussion about cervical cancer prevention, free HPV testing, relatively large numbers of attendees, education of the woman concerning cervical cancer and HPV and they can now take this information back to their communities, access to self sampling technique, access to same day oncHPV results, and a same day test and treat strategy is needed, one stop location for HPV vaccination for age appropriate children. Weaknesses of this mother-child campaign include the long wait times for same day results, once the campaign is over there will be a cost for women who present for HPV testing (but not vaccination of age appropriate children), long waiting time for test re-sults (2–4 weeks) especially when the AmpFire machine is no long at Meskine, thus risk of loss to follow-up of positive women.

## 5. Conclusions

The mother–child approach to cervical cancer prevention benefits the women who come for screening with both education and testing. It also benefits their progeny, who are vaccinated. There are elements of such campaigns that lead to success, such as buy-in from local leaders, extensive sensitization and education, and free access to healthcare in oppressed populations like the geographically isolated, uneducated, or low-educated

(Figure 8). Currently, such approaches have been funded by grants or donors. One cannot highlight enough the massive need for universal healthcare coverage for evidence-based prevention programs like cervical cancer screening and vaccination in LMIC countries.

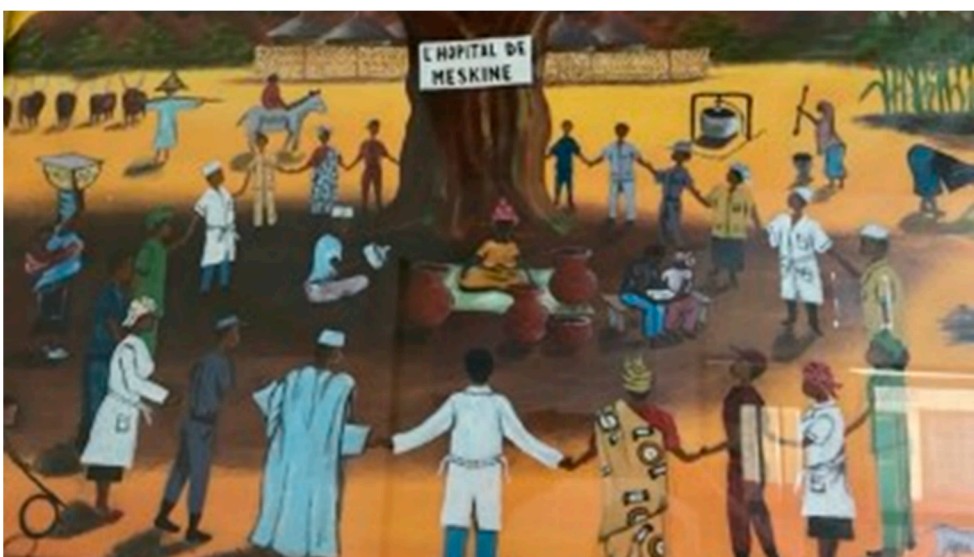

**Figure 8.** Painting at Meskine demonstrating that it is all about partnerships.

**Author Contributions:** Conceptualization, L.E., F.M., L.K., B.S., M.S., E.W. and T.W.; methodology, L.E., F.M., E.W. and T.W.; software F.M., E.W. and T.W.; validation, F.M. and T.W.; formal analysis, L.E., F.M. and T.W.; investigation, L.E., F.M., B.S., M.S., E.B., E.V. and T.W.; resources, L.E., F.M., E.W. and T.W.; data curation, L.E., F.M., L.K., B.S., E.B., E.V. and A.F.; writing—original draft preparation, L.E.; writing—review and editing, F.M., L.K., B.S., M.S., E.B., E.V., A.F., E.W. and T.W.; visualization, L.E., F.M. and L.K.; supervision, F.M.; project administration, F.M., B.S. and M.S.; funding acquisition, L.E., F.M., E.W. and T.W. All authors have read and agreed to the published version of the manuscript.

**Funding:** This research received no external funding.

**Institutional Review Board Statement:** Ethical review and approval were waived for this study as it is a retrospective assessment of an implemented program. The Ethics committee did review the final manuscript and photographs.

**Informed Consent Statement:** Patients' verbal consent was provided for the screening program.

**Data Availability Statement:** The data presented in this study is available on request from the corresponding author.

**Conflicts of Interest:** The authors declare no conflict of interest.

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
