# Peer review of "Mother–Child Approach to Cervical Cancer Prevention in a Low Resource Setting: The Cameroon Baptist Convention Health Services Story†"

_curroncol, doi:10.3390/curroncol31060244_

Round 1

Reviewer 1 Report

Comments and Suggestions for Authors

Dear authors,

thanks for this interesting paper on a novel approach to raise awareness and improve HPV vaccination and prevention in LMIC countries.

This provides nice addition to literature and should merit publication.

Please check correct spelling throughout manuscript (e.g. VILLI)

Please provide also some lines about vaginal HPV related lesions, and awareness should be raised (https://doi.org/10.1002/jmv.29474)

Comments on the Quality of English Language

Minor

Author Response

We thank you and your reviewers for your helpful comments. Amendments have been made as outlined below.

Reviewer 1

  1. Check spelling mistakes like VILLI.

VILLI has been revised to VILI throughout the manuscript.

  1. Add information on vaginal HPV related lesions.

The reference for vaginal HPV was reviewed. Given the length of the current manuscript and the relatively small part vaginal disease plays in the context of an intact cervix, we respectfully did not add any information about vaginal HPV related lesions.

Reviewer 2 Report

Comments and Suggestions for Authors

Dear Authors,

Your paper describes very important public health issue. However the article is very narrative. I would suggest adding some statistical analyses to assess the profile of HPV positive women, and to compare them to HPV-negative.

Your paper needs a thorough language editing as well.

Please avoid abbreviations like yo; you should decide to name the approach either “mother-child” or “mother-and-child” and be consistent for the entire article. The term “healthcare” is more appropriate than “health care”. It seems that a preposition is missing in the first sentence of the abstract. The next sentence makes the impression that only the daughters are vaccinated but sons were not. A noun is missing within the next sentence of the Abstract. COVID-19 should be entirely capitalized.

 It would be better if you also add the number of the positive cases that were found within the abstract.

Page 2, lines 65 and 66: a space should be added after the last number

Page 3, lines 104 and 110: a space is missing after Jan. In addition, please do not abbreviate the months. Line 110: it is in American English. Please use either British or American but do not mix them.

Page 6, Table 1. Since the majority of the women were 30-49, I would suggest to split their group and to show the number and % of 30-39 and 40-49. The other groups intervals should be consistent: instead of 50-60, you could add 50-59 and 60-69. There is a gap in the table, no woman is between 61 and 70 so please add them (if there are any). The last group should be 70+ (instead of 71+).

There are 9 women that did not report their age as well as 9 did not report their marital status. Are they the same? If yes, then how did you collect the other questions’ responses bud missed these two?

What does it mean <5 HIV positive? Please specify the exact number. I guess it is 4.

You could simplify the table by decreasing the decimals to 1 and remove the symbol % as it is written in the column title.

Page 6, line 163. The word “women” is omitted.

Lines 163-166: there is something wrong with the calculations. The number of HPV+ and HPV- is 500 when add both presented numbers. What about the rest 5 results? In addition you calculated 74 out of 105 as 73.3% but it is 70.5? I would advice to report these results in a table. Then you could further analyse the differences in HPV+ and HPV- cases in their age, education etc. demographics.

Page 9, Table 2 reported 506 screened women? Were they 505? Here you could add a statistical test comparing the proportion of HPV+ in these two settings.

Table 2 was not mentioned in the text. 

Comments on the Quality of English Language

Please avoid abbreviations like yo; you should decide to name the approach either “mother-child” or “mother-and-child” and be consistent for the entire article. The term “healthcare” is more appropriate than “health care”. It seems that a preposition is missing in the first sentence of the abstract. The next sentence makes the impression that only the daughters are vaccinated but sons were not. A noun is missing within the next sentence of the Abstract. COVID-19 should be entirely capitalized. 

The entire paper needs language editing. 

Author Response

We thank you and your reviewers for your helpful comments. Amendments have been made as outlined below.

Reviewer 2

  1. The reviewer asked for statistical analysis.

Given the short timeframe we were given to return these amendments, the analysis requested was not feasible.

  1. Editing requested

Abbreviation yo – has been changed to years old throughout the document.

Mother and child has been changed to Mother-child consistently through the document.

Health care has been changed to healthcare.

COVID-19 has been capitalized in the document.

  1. Add number of positive cases in the abstract

The abstract now has the total number and percentages.

  1. Space at line 65-66

Changes were made.

  1. Space after Jan

Date has been expanded.

  1. Groupings in table 1

I apologize for the error. It should read 50-69 not 50-60. This has been edited.

  1. The 9 women without and age and the 9 without a marital status.

These are not necessarily the same women. All of the questionnaires were completed by a nurse or physician who interviewed the woman. Some of the tribal languages were not represented by the healthcare interviewers. When this occurred and where possible we tried to interview by including a family member who could speak one of the languages of the interviewers. Birth certificates do not exist for many in this population.

  1. Less than <5

Convention is not to list the cell number if it is less than 5.

  1. Table 1 Percent column

The column has been amended as requested (ie., remove the % sign and only report to one decimal place).

  1. The word woman

The word woman was added after 505.

  1. Calculations

We agree with the reviewer that the 74/105 is 70.5% and not 73.3%. This has been amended in the abstract and the manuscript.

We are not sure where the reviewed got the sample size number of 500 as the manuscript repeatedly refers to 505.

We agree there was a typo in Table 2 and this has been amended.

  1. Table 2 was not mentioned in the text.

Table 2 is now referred to in the text just before reference [16].

Reviewer 3 Report

Comments and Suggestions for Authors

The data reported in the manuscript are of interest for implementing the  prevention of cervical cancer in LMIC in order to reach the "elimination cervical cancer goal" set by WHO. In this study, a mother-child combined approach was used to provide HPV-based screening (by AmpFire HPV assay) to the mothers and HPV vaccination to their 9-14-yo children in a Far North Region of Cameroon (that has a high rate of poverty). A grant of the Charities Aid Foundation of America covered the cost of breast and cervical cancer screening for 500 women, while the quadrivalent HPV vaccines were provided through the EPI program. Group or individual education and training on how to perform the self-sampling procedure were provided by local staff. Overall, during the 5-days campaign, 500 women (of whom, 21% were HPV-positive) were screened and 92 (58 girls and 34 boys) were vaccinated. All the 105 women testing HPV-positive underwent visual inspection with acetic acid (VIA) or Lugol's iodine (VILI); the 8 resulted VIA/VILI positive were treated by thermocoagulation or LEEP. The results confirm that the mother-child approach is feasible and has a good acceptance by the target population. The manuscript needs some revision to implement methodological aspects, highlight strenghts and limitations of the protocol, and improve the quality of English language (see details in the specific box).

MAJOR COMMENTS:

-INTRODUCTION, lines 70-80: of the two references reported in line 73, I could not find the report indicated in ref. 7, while the abstract reported in ref. 8, refer to a project based on a mother-daughter approach conducted in Cameroon in 2022 that involved 1691 girls and 827 women. Please, provide additional data on the report and on the project.

-METHODS, lines 100-115: please, explain how the protocol was defined; in particular, how did the women get involved (press release in French doesn't seem appropriate, according to what reported in lines 275-277) and the role of the local religious leaders and why the campaign lasted only 5 days.

-ETHICS, lines 152-153: besides the final manuscript, was the protocol reviewed?

-RESULTS, lines 155-161: an overview of the education and training phases (i.e., how many groups/sessions were done; what was performed in the 5-days campaign and what in the previous and/or following days; problems/failures occurred) can add useful information for both understanding the challenges and contextualize the results.

-HPV vaccination: the number of vaccinated children is reported as 91, but it doesn't match with the sum of "34 boys and 58 girls".

-DISCUSSION, lines 246-262: a more detailed comparison of the results of this project in comparison to the results of the project described in ref. 8 is necessary.

-DISCUSSION: please, add a paragraph on strenghts and limitations of the study.

Comments on the Quality of English Language

Some sentences (i.e., lines 50-53, lines 163-166, lines 232-237) are not clear and need to be rephrased.

Some words or acronyms need to be corrected: VILI (instead of VILLI or vili); TogetHER (instead of ToGETHER).

Line 50: "...HPV self test..." should be corrected as "...HPV self sampling...".

Typing errors are present throughout the manuscript.

Author Response

We thank you and your reviewers for your helpful comments. Amendments have been made as outlined below.

Reviewer 3

  1. References 7 and 8

We agree that ref 7 is to a report submitted to the Together Foundation and is not in the published literature.

Ref 8 is only a meeting abstract. There is no published manuscript that we are aware of.

  1. Line 105-115 do not align with lines 275-277

We amended lines 105-115 to indicate that only one school had been sensitized as reflected in lines 275-277

“WHP provided education of community and religious leaders which included sensitization of local churches and mosques, a school, and village meeting centers (see Figure 1).”

The mass campaign lasted only 5 days but women and/or children could attend the hospital for screening or vaccination respectively.

  1. Protocol review

There was no protocol for this work. It was a mother-child campaign. Thus ethics only reviewed the final manuscript which was much more detailed than a campaign report.

  1. Number of group sessions

This information was not captured. In part, larger group sessions occurred earlier in the day when there were large crowds that gathered. As the day unfolded, smaller and smaller numbers of women gathered for teaching sessions which were provided to new attendees in the language of their preference and provided one of our nurses who could speak that language.

Each day unfolded the same: women gathered – they were taught about cervical cancer and cervical cancer prevention – they were informed about HPV self testing – they individually were interviewed for demographic information – they were provided a self test if they signed consent – they conducted their own self test in the washroom – the self tests were batched and analyzed – women had an individual breast exam by a health care provider – they were given their self test HPV result – those with negative results were discharged and those with positive results went on the further assessment on the same day.

  1. 34 plus 58 is 92 not 91

The reviewer is correct, and this has been amended throughout the document.

  1. Detailed comparison of this project and reference 8 is not feasible as there are no details available for ref 8.
  2. Strengths and weaknesses

The original lines 268-309 are quite detailed about the uniqueness, the issues/weaknesses and strengths of the program. A further paragraph has been added as requested.

  1. Spelling mistakes (as described) have been corrected.
  2. Lines 50-53 have been reworded as follows:

More recently HPV self sampling or physician acquired testing is a highly sensitive screening test (more than 90%). However, a positive result for High risk HPV type(s) mainly indicate current infection. Women must undergo a second test like VIA/VILI to determine if there is infection and dysplasia present.

  1. Lines 163-166 has been re-written as follows:

505 women completed HPV tests. The HPV test was negative in 404 (80%). Of the 101 positive HPV tests (20%): there were 9 with HPV 16 (9%), 11 with HPV 18 (10.1%), 74 (73%) with HPV other 13 high risk types. Seven percent had more than one oncHPV type: 3 with HPV 16 and HPV other 13 types (3.0%), 2 with HPV 18 and HPV other 13 types (2.0%), 1 with HPV 16 and 18 (1.0%), 1 with HPV 16, 18 and other 13 HR types (1.0%).

  1. Lines 232-237 has been reworded to:

The literature addresses effective ways to optimize cervical screening attendance in LMIC. In 24 studies (15 RCT and 9 cluster with RCTs including 1 study from Cameroon) self sampling was superior to VIA (RR 1.93 95% CI 1.66-2.25); reminder phone calls and SMS tests facilitated attendance when compared to letter invitation (RR 1.72, 95%CI 1.27-2.32, RR 1.59, 95% CI 1.46-5.19); free or subsidized cost optimized attendance (RR 1.60, 95% CI 1.10-2.33); and community-based HPV testing was superior to hospital collected HPV testing (RR 1.67, 95% CI 1.53-1.82) [15].

Round 2

Reviewer 2 Report

Comments and Suggestions for Authors

Dear Authors,

A great work has been done and your manuscript is significantly improved!

One “yo” remained in the Abstract so please remove it.

Narrative comparison is not a scientific approach. It is important to identify significant factors leading to HPV infection among the target population.

I appreciate your efforts to screen the population and to help them get adequate treatment! I know that Cameroon needs this paper to be published. 

Author Response

Re: Curroncol-2978837

Mother and Child approach to cervical cancer prevention in a low resource setting: The Cameroon Baptist Convention Health Services story.

We thank you and your reviewers for your helpful comments. Amendments have been made as outlined below.

Reviewer 1

  1. Abstract contains – yo

This has been revised to read “years old”

Reviewer 3 Report

Comments and Suggestions for Authors

The authors have answered to my comments, but the reasons provided and/or the changes to the text are not always adequate. They say that no protocol was prepared for this campaign, and no record of the number of the sessions to inform and train the women was kept. While ackowledging that this mother-child campaign was successful, the collection of these informations would have been useful to gain indications for implementing future campaigns  (i.e., among the difficulties encountered, these data are important: women's feedback on the sessions of information and training; number of staff speaking local languages needed; need of battery back up for keeping the HPV testing machine in function in case of power failure).

Moreover, while understanding that the references 7, 8 and 16 are not accessible, I would recommend to include in the text the informations available to the authors that prompted them to refer to theses documents.

The correct description of the screening strategy is that "the women use a self-sampling device to obtain the cells that are then tested for HPV by healthcare professionals". THE WOMEN DO NOT COLLECT HPV SAMPLES. To this aspect, corrections to the text need to be done: as an example, line 27 of the ABSTRACT should read "to having each woman retrieve a personal sample. Self-collected samples were analysed for HPV the same day. All"  [other corrections also in lines 51-52, and 120]

Comments on the Quality of English Language

Some revision of English language is still necessary, for both better construction of some sentences and correction of typos (i.e., line 56: nanovalent should read nonavalent).

Author Response

Re: Curroncol-2978837

Mother and Child approach to cervical cancer prevention in a low resource setting: The Cameroon Baptist Convention Health Services story.

We thank you and your reviewers for your helpful comments. Amendments have been made as outlined below

Reviewer 2

  1. Reviewer 2 appreciates that References 7,8 and 16 are abstracts or project reports so peer reviewed publications on these projects are not available.

As requested we have added information in the manuscript concerning why we illude to these reports.

Under the section – uniqueness of the mother child approach: we highite the aspects of the approach in two prior projects [7,8]. Again in the manuscript we now indicate that aspects of the [16] project informed the conduct of the current work reported on in Meskine.

  1. We have made the recommended word change: The women use a self-sampling device to obtain the cells that are then tested for HPV by healthcare professionals.
  2. Correct spelling of nanovalent

This has been correct to nonavent through out the document.

  1. English language and grammar.

Manuscript has been reviewed by 2 physicians where english is their first language. Edits have been made for language use and grammar.

Respectfully resubmitted

Round 3

Reviewer 2 Report

Comments and Suggestions for Authors

Great! 

Ready to be published.